# The Impact of COVID-19 on Epidemiological Features of Spinal Cord Injury in Wuhan, China: A Comparative Study in Different Time Periods

**DOI:** 10.3390/medicina59101699

**Published:** 2023-09-22

**Authors:** Ruba Altahla, Jamal Alshorman, Xu Tao

**Affiliations:** 1Department of Rehabilitation, Tongji Hospital, Tongji Medical College, Huazhong University of Science and Technology, Wuhan 430030, China; i201922121@hust.edu.cn; 2Department of Orthopedics, Wuhan Union Hospital, Tongji Medical College, Huazhong University of Science and Technology, Wuhan 430030, China; jamalking61@yahoo.com

**Keywords:** spinal cord injury, COVID-19, non-COVID-19, rehabilitation, comparative study

## Abstract

*Background and Objectives:* Spinal cord injury (SCI) is a severe affliction that can have a profound impact on a person’s ability to move and feel, affecting a significant number of individuals. However, rehabilitation after SCI treatment remains a critical method to improve motor–sensory functions, which improves the patient’s quality of life. This study aims to describe the epidemiological profile of SCI during the COVID-19 pandemic (“COVID-19 period”) and before and after the COVID-19 pandemic (“non-COVID-19 period”) in Wuhan City, Hubei Province, China. *Materials and Methods*: Medical records of 93 patients diagnosed with SCI admitted to the rehabilitation department of Wuhan Tongji Hospital from January 2019 to May 2023 were retrospectively reviewed. Basic demographics and clinical characteristics such as level of injury, American Spinal Injury Association (ASIA) Impairment Scale, treatment method, and concomitant injuries were analyzed. *Results*: Forty patients with SCI from the non-COVID-19 period and fifty-three patients from the COVID-19 period were identified. The mean ages were 38.80 ± 17.71 and 44.53 ± 13.27 years, respectively, with a consistent male-to-female ratio of 2:1 across both periods. Notably, falls accounted for the most prevalent mechanism of injury, constituting 50% of cases during the non-COVID-19 period and 37.74% during the COVID-19 period. The most common initial ASIA grade was B in the non-COVID-19 period and grade C in the COVID-19 period. In addition, the final ASIA grade after treatment was grade C in the non-COVID-19 period and grade D in the COVID-19 period. *Conclusions*: A greater proportion of males suffer from SCI, and the primary causes are falls and traffic accidents. Workers are the most vulnerable group to SCI among all patients. Prevention strategies should be customized based on the unique characteristics of SCI patients. This study highlights the importance of SCI rehabilitation.

## 1. Introduction

Spinal cord injury (SCI) is a severe ailment that can significantly diminish a person’s quality of life (QOL). It can be the result of either traumatic events such as motor vehicle accidents (MVAs), falls, or sports injuries, or non-traumatic causes such as infections or tumors. The COVID-19 pandemic has added another layer of complexity to the management of SCI, as patients with this condition may be at higher risk for severe illness and complications from the virus [1,2]. Data on the occurrence of SCI worldwide before and after the COVID-19 pandemic are limited. Nonetheless, it was estimated that the incidence rate of traumatic SCI in China was 23.1 cases per million population per year before the pandemic [1,3]. In other countries, the incidence rates of SCI widely vary. According to data from the National Spinal Cord Injury Statistical Center in the United States, the estimated annual incidence of SCI is approximately 17,700 new cases per year or about 54 cases per million population [4,5]. However, it is difficult to make a direct comparison of incidence rates between countries due to differences in the populations being studied and other factors. Due to limited research, the impact of the COVID-19 pandemic on SCI incidence rates is not well established. Nevertheless, another study claimed a reduction in the occurrence of traumatic SCI in Italy during the COVID-19 period compared to another period [6].

Despite the challenges arising from the COVID-19 pandemic, rehabilitation remains a critical component of care after SCI treatment. Rehabilitation can improve motor–sensory function, QOL, and prevent long-term complications, such as pressure ulcers and musculoskeletal complications [3]. In recent years, there has been a rise in new techniques in SCI rehabilitation [7,8], such as new research on the use of robotic technology in rehabilitation departments after SCI treatment [7,8,9,10,11], as well as the cooperation of multidisciplinary teams in providing high-quality care to the SCI patients [12]. Wuhan City was the first city in the world to experience an outbreak of COVID-19, and the pandemic had a significant effect on the healthcare system and treatment methods. The purpose of this study is to show the impact of traumatic and non-traumatic SCI (TSCI and NTSCI) on patients during the COVID-19 pandemic, as well as before and after the pandemic, highlight the importance of rehabilitation methods in the management of SCI, and discuss the challenges presented by COVID-19 in the context of rehabilitation after SCI.

## 2. Methodology

As there was no community-based registration system for TSCI or NTSCI established in Wuhan City during the two different periods (the first before and after the COVID-19 outbreak and the second during the COVID-19 outbreak), the patients’ data obtained for this study were gathered from medical records after exact diagnosis with either traumatic spinal cord injury (TSCI) or non-traumatic spinal cord injury (NTSCI) according to the Affiliated Tongji College Hospital during the specific time periods (denoted hereafter as the “non-COVID-19” and “COVID-19” periods). The non-COVID-19 period covered 2019 and 2023, while the COVID-19 period covered 2020–2022. Moreover, in this study, the International Classification of Disease Version 10 and the diagnostic code of traumatic spinal cord injury (TSCI) and non-traumatic spinal cord injury (NTSCI) were employed.

### 2.1. Ethical Considerations

This study was approved by the ethical committee of the Affiliated Tongji Medical College Hospital (ethical approval number: TJ-IRB20210314).

### 2.2. Data Collection and Statistical Analysis

The methodology used for data collection in this study involved a review of the medical history of patients who met the inclusion and exclusion criteria. The patients’ records provided demographic characteristics, including age, gender, marital status, education level, work type, ethnicity, and the specific mechanism of injury (MOI): traumatic (vehicle accidents, fall, and sport injury) or non-traumatic (disc herniation, myelopathy/myelitis, spinal cord ischemia, and tumor) [13]. In addition, the medical records from December 2019 to April 2023 included information on injury severity, as measured by the American Spinal Injury Association (ASIA) scale, a standardized test that evaluates the extent of spinal cord injuries. It assesses motor function based on specific muscle groups and sensory function based on specific areas of the body, and it also includes an examination of the anus and rectum. The results of these tests determine the severity and location of the injury [14], injury level (cervical, thoracic, lumbar, and thoracolumbar), lesion completeness (complete and incomplete), treatment method (surgical and conservative), and paralysis type (paraplegia and tetraplegia). Moreover, in this study, we have used the International Spinal Cord Injury Core Dataset version 1.1.

### 2.3. Inclusion and Exclusion Criteria

The inclusion criteria for this study were SCI patients who were admitted to the hospital within the specified time from December 2019 to April 2023 and transferred to the rehabilitation department, treated during the non-COVID-19 period (2019 and 2023) and the COVID-19 period (2020–2022). Additionally, other patients included in this study were those who received a diagnosis of SCI based on the International Standards for Neurological Classification of Spinal Cord Injury (ISNCSCI). However, we excluded patients with incomplete medical records; patients with pre-existing medical conditions, such as heart disease or brain tumor; patients with a history of drug or alcohol abuse; patients with concurrent injuries, such as traumatic brain injury or extremities fracture; patients with intervertebral disc disease or spinal fractures without SCI; patients with an unclear diagnosis in their medical records; individuals who had suffered fatal injuries but were never admitted to the hospital, and patients who did not transfer to the rehabilitation department of Tongji Hospital. Similar to previous research, the participants in this study were stratified into distinct age groups [15]. This study recorded marital status as either married or unmarried, while the MOI was categorized as either a road traffic accident (RTA) (involving a bicycle, car, motorbike, or fall) or other causes [16]. Moreover, injuries caused by falling objects, machinery-related injuries, sports injuries (dance), and NTSCI were due to disc herniation, myelopathy, myelitis, spinal cord ischemia, and spinal cord tumors. Occupations included driver, farmer, government office, kindergarten student, retired, student, teacher, worker, and others. The medical imaging records were used to determine the neurological level of injury, which included the cervical, thoracic, lumbar, and thoracolumbar segments.

### 2.4. Statistical Analysis

The statistical analysis was carried out using a particular software (SPSS version 23.0). Descriptive statistics were used to present the characteristics of the study population. However, percentages, means, and standard deviations described the demographic characteristics of the patients, such as age, gender, and education level. This study also used frequency distributions to describe the distribution of injury severity, such as the ASIA classification and injury level, across different subgroups of patients, such as traumatic versus non-traumatic SCI. Furthermore, statistical tests, such as chi-square tests, were used to investigate the relationships between various variables, such as the ASIA classification and injury level based on MOI or work type.

## 3. Results

A total of 162 cases of spinal cord injury (SCI) were initially considered for this study, from 2019 to 2023. However, during the screening process, 69 cases were excluded for various reasons. These reasons included 22 cases with traumatic brain injury, 16 cases with spinal column fractures but no SCI, 13 cases with heart disease, 15 cases with unclear diagnoses, and 19 cases with incomplete medical records or uncertain diagnoses. The yearly distribution of patients was 16 patients (17%) in 2019, 11 patients (12%) in 2020, 24 patients (26%) in 2021, 18 patients (19%) in 2022, and 24 patients (26%) in 2023 (Figure 1). Thus, 93 patients after SCI and rehabilitation between December 2019 and April 2023 were included in the final analysis. Moreover, 40 patients were in the non-COVID-19 period of 2019 and 2023, while 53 COVID-19 patients presented in 2020–2022 (Figure 2).

### 3.1. COVID-19 and Non COVID-19 Periods

The demographic features of the COVID-19 and non-COVID-19 SCI patients are shown in Table 1. The mean age of the 40 SCI patients collected from the non-COVID-19 period was 38.80 ± 17.71 years. However, the male-to-female ratio was 2:1. Most patients (26 (65%)) were married, and their education level was mainly high school (10 (25%)). The most frequent occupation was a worker, while teacher and retired were the least frequent. Ethnicity was mostly Han, for 39 (97.5%) patients. However, NTSCI was found in 9 (22.55%) patients and TSCI in 31 (77.5%). The most common ASIA grade was B, and the most common injury level was in the cervical region. MOI was divided into nine categories, and fall was the most common. Thirty-four (85%) patients required surgical treatment methods, while six (15%) underwent conservative treatment (Table 1). In the final follow-up, the final ASIA was improved in many patients (Figure 3), and seven (17.5%) patients showed complete recovery.

A total of 53 SCI patients were recruited from the COVID-19 period according to the inclusion and exclusion criteria. The mean age was 44.53 ± 13.27 years, and the male-to-female ratio was 2:1. Most patients (43 (81.13%)) were married, and the most common education level was college (23 (43.4%)).

Work type was divided into nine categories, which showed the most frequent occupation was a worker. Ethnicity was mostly Han, for 45 (84.91%) patients. Moreover, NTSCI was seen in 18 (33.96%) patients, and TSCI in 35 (66.04%). The most common initial ASIA was grade C, in 20 patients (37.74%). The injury level mainly presented in the thoracic region. MOI was divided into nine categories, and fall was the most common type. Forty-four (83.02%) patients required surgical treatment methods, while nine (16.98%) underwent conservative treatment. In the final follow-up, the final ASIA was improved in many patients (Figure 1), and 11 (20.75%) patients showed complete recovery.

### 3.2. Level of Injury

This study found a bimodal distribution of SCI levels, as illustrated in Table 2. Injury to the cervical region formed the first peak, followed by the thoracic region in the non-COVID-19 period, while in the COVID-19 period, the thoracic region comprises the first peak, followed by the cervical region (Table 1). The different levels of injury were associated with different occupations. Most people with cervical, thoracic, and thoracolumbar area injuries, 13 (34.21%), 10 (33.33%), and 4 (33.33%), respectively, were workers. In terms of education level, the largest group of patients had completed college, followed by high school and no formal education (Table 2). There was no statistically significant difference in the distribution of injury levels across the different education levels (*p* = 0.943). There was, however, a statistically significant difference in the distribution of the initial ASIA classification across the different injury levels (*p* = 0.009). In particular, patients with cervical injuries were more likely to have an initial ASIA classification of A or B, while patients with thoracic or thoracolumbar injuries were more likely to have an initial ASIA classification of C or D (Table 2). However, comparing the TSCI and NTSCI with the ASIA scale, ASIA C was the most common grade in NTSCI, followed by grades B, D, and A, respectively. However, in TSCI, the most common ASIA grade was B, followed by A, C, and D, respectively (Figure 3).

### 3.3. ASIA and Mechanism of Injury

The most common MOI was fall, followed by car accidents and myelitis. There was no statistically significant difference in the distribution of injury levels across different mechanisms of injury (*p* = 0.596) (Table 2). In terms of initial ASIA grades, the largest group of patients had grade B, followed by grades A and C (Table 2), and only a few patients had grade D. There was a statistically significant difference in the distribution of the initial ASIA classification across different MOIs (Figure 4). Patients with fall-related injuries were more likely to have a lower initial ASIA classification, while patients with dance-related injuries were more likely to have a higher initial ASIA classification. However, when examining MOI, the most common cause of injury was falls, followed by myelitis and motorbike accidents (Figure 4).

### 3.4. Age and Mechanism of Injury

Among patients with bicycle-related injuries, the majority were in the 39–58 age group, while the remaining patients were in the 59–72 age group. For car-related injuries, the highest number of patients were in the 19–38 age group, followed by the 39–58 age group. Dance-related injuries were only reported in the 0–18 age group. Patients with disc herniation-related injuries were mainly in the 39–58 age group. However, for fall-related injuries, majority of patients were in the 39–58 age group, followed by the 19–38 age group. Patients with motorbike-related injuries were mainly in the 39–58 age group, followed by the 19–38 age group. Myelitis-related injuries were distributed across all age groups, with the highest number of patients in the 39–58 age group. Patients with spinal cord ischemia-related injuries were mainly in the 39–58 age group. Tumor-related injuries were mainly reported in the 0–18 and 39–58 age groups. It is worth noting that the sample size for some age groups and MOI was quite small, which limits the generalizability of the results (Table 3 and Figure 5).

### 3.5. ASIA Scale with Gender

The distribution of ASIA grades among patients with SCI ws based on their gender. Overall, there was no statistically significant difference in the distribution of ASIA grades between male and female patients (*p* = 0.066). However, there were some differences in the proportions of ASIA grades between the two genders. In particular, male patients had a slightly higher proportion of ASIA grades B and C, while female patients had a slightly higher proportion of ASIA grade D (Table 4).

### 3.6. Treatment of SCI and Disturbances of Initial and Final ASIA among SCI Individuals

The distribution of the final ASIA classification among patients with SCI was based on their treatment type (conservative or surgical). Patients with a final ASIA classification of A, B, C, and D were more likely to undergo surgical treatment compared to conservative treatment (*p* = 0.001) (Table 5). In particular, patients with final ASIA grades of A, B, and C had a higher proportion of surgical treatment than conservative treatment, while patients with a final ASIA grade of D had a higher proportion of conservative treatment. On the other hand, the percentage of patients’ improvement to grade E was higher in the conservative treatment than the surgical treatment group (Table 5). However, in the final follow-up, the final ASIA was improved in many patients (Figure 6), and 11 (20.75%) patients progressed to complete recovery (Table 1).

## 4. Discussion

A recent systematic review of 17 studies conducted in China revealed that the epidemiological features of SCI differed across different regions [17,18]. This implies that tailored preventive measures must be implemented based on the specific characteristics of each region. This retrospective, cross-sectional study of SCI in Wuhan, Hubei, China, from 2019 to 2023 aimed to describe the demographic and clinical characteristics of patients with SCI during both non-COVID-19 and COVID-19 periods. Since this study was conducted retrospectively, it was inevitable that some data might have been missing. However, efforts were made to minimize data loss by thoroughly examining all relevant medical records to ensure that the resulting dataset was as comprehensive as possible. Our findings showed differences in the distribution of age, gender, education level, work type, and MOI between the two groups.

In the non-COVID-19 group, the mean age was younger (38.80 ± 17.71 years) compared to the COVID-19 group (44.53 ± 13.27 years). These findings are in line with a recent study conducted in Northwest China, which indicated that most patients were between the ages of 30 and 39 [15]. This could be attributed to the fact that older adults might have had increased risk factors for traumatic events during the COVID-19 pandemic, such as decreased physical activity, limited access to healthcare, and mental health issues. Moreover, the pandemic might have disproportionately affected older individuals with pre-existing neurological conditions, leading to an increased incidence of NTSCI [19]. Regarding gender, both groups had a higher proportion of male patients, with a male-to-female ratio of roughly 2:1, which is consistent with previous studies reporting a male predominance in SCI [18,19,20,21]. This could be related to the higher involvement of males in occupations and activities that increase the risk of traumatic events [22].

The most prevalent occupational group in this study was workers, with 12 (30%) and 16 (30.19%) in each group. This finding is in line with the previous literature, as individuals in manual labor occupations are more likely to experience traumatic events leading to SCI [23]. The distribution of occupational groups was not significantly different between the two groups, which could be explained by the fact that the pandemic had a widespread impact on various occupations, not just manual labor. Our findings showed that the proportion of married patients was higher than that of unmarried patients, which could be because most patients were middle-aged, which is when most Chinese people marry.

In terms of education levels, college education was more prevalent in the COVID-19 group, while high school education was more common in the non-COVID-19 group. This might suggest that higher education levels could be associated with an increased awareness of personal safety and risk factors during the pandemic.

The most common MOI in both groups was fall, with 20 (50%) and 20 (37.74%) patients, followed by car accidents with 7 (17.5%) and 11 (20.75%), respectively. This is consistent with previous studies that reported falls as the leading cause of SCI globally [24]. The COVID-19 pandemic might have contributed to an increased number of falls due to factors such as reduced physical activity, isolation, and mental health issues [19].

In most developed countries, traffic accidents have traditionally been the primary cause of SCI, whereas the most TSCI resulting from high falls have occurred in the construction industry [21,25,26].

This trend can be explained by China’s rapid industrialization and the increasing number of large-scale infrastructure projects. As a result, there is a pressing need to focus on developing effective measures to prevent falls in the construction industry. Various factors, such as falls (high and low), traffic accidents, impact with falling objects, sports, and violence, can lead to SCI, and the occurrence of these factors can differ depending on the country and region. According to a survey on the epidemiology of SCI carried out in Canada, traffic accidents were identified as the primary cause of these injuries in 2006. However, a later survey conducted in 2009 indicated that falls (both high and low) had taken over as the leading cause of SCI, replacing traffic accidents [27,28]. In seven countries located in the Middle East and North Africa regions, a survey found that traffic accidents were still the leading cause of SCI, followed by falls (both high and low), violence, and sports [29]. The analysis of injury locations in this study demonstrated two noticeable peaks, which align with the findings of previous studies. Before the COVID-19 pandemic, the first peak was observed in the cervical region, whereas during the COVID-19 pandemic [16,27], the first peak shifted to injuries in the thoracic area. Previous research has revealed that most SCIs, ranging from 55% to 75% of all cases, involve cervical injuries, making them the most prevalent type of SCI [28,29].

The reason why cervical injuries are more common than other types of SCI could be attributed to the fact that the cervical vertebrae have lower mechanical stability, which makes them more vulnerable to damage. Our results showed a statistically significant difference in the distribution of the initial ASIA classification across different injury levels [20]. This finding is consistent with the previous literature, indicating that cervical injuries are more likely to result in severe neurological impairment [30]. Additionally, there were no significant differences in the distribution of the initial ASIA classification across different MOIs, which could be due to the wide variety of MOIs observed in this study.

During the non-COVID-19 period, ASIA grades A and B comprised most SCI cases. During the COVID-19 period, most spinal cord injury (SCI) cases were classified as ASIA B and C injuries in Asia. Typically, a complete SCI of grade A is the result of injuries from traffic accidents and falls from great heights. Conversely, incomplete SCIs, mainly of grade B, tend to occur more frequently in falls from lower heights. Studies by Williams et al. and Thietje et al. [31,32] have revealed that patients with grade A are more susceptible to developing depressive disorders and exhibiting suicidal behavior. Therefore, healthcare providers and family members should provide additional care and support to these patients to prevent depression-related suicide. Furthermore, this study found that for both periods, incomplete SCI was more frequent than complete injuries.

Even though incomplete SCIs are more frequent than complete injuries, it is still crucial to create first aid technologies that can offer timely and appropriate treatment to individuals who have suffered an SCI.

The final ASIA classification showed improvement in many patients in both groups, which highlights the importance of timely and appropriate rehabilitation and management of SCI patients [20,33]. This finding aligns with previous studies that have reported positive outcomes following rehabilitation [34]. This study revealed an association between the final ASIA scale and the treatment method. The patients with a final ASIA classification of A, B, or C had a higher proportion of surgical treatment than conservative treatment, while patients with a final ASIA classification of D had a higher proportion of conservative treatment.

Along with an association between the ASIA and MOI, there was a statistically significant difference in the distribution of the initial ASIA classification across different MOIs. Patients with fall-related injuries were more likely to have a lower initial ASIA classification, while patients with dance-related injuries were more likely to have a higher initial ASIA classification.

Moreover, we found an association between age and the mechanism of SCI fall-related injuries: most patients were in the 39–58 age group, followed by the 19–38 age group, while for car-related injuries, the highest number of patients was in the 19–38 age group, followed by the 39–58 age group. Despite the association between the ASIA scale and gender, there were some differences in the distribution of the ASIA classification between male and female patients. The overall statistical analysis did not show a significant difference (*p* = 0.066) between the two groups.

Rehabilitation is a crucial component of the management of patients with SCI. It can aid in functional recovery, prevent complications, and improve long-term outcomes. Multidisciplinary rehabilitation programs that include physical therapy, occupational therapy, and psychological counseling are effective in promoting recovery in SCI patients. Studies conducted in 2020 and 2021 have demonstrated that early and intensive rehabilitation can lead to better outcomes in terms of functional improvement and QOL for SCI patients [33,35]. Furthermore, emerging technologies, such as virtual reality and robotics, have demonstrated promise in enhancing rehabilitation outcomes for SCI patients [36]. Overall, rehabilitation is a vital aspect of the management of SCI patients, and healthcare providers should ensure that patients receive timely and appropriate rehabilitation services.

Failure to participate in rehabilitation after a SCI can lead to long-term disability and a reduced QOL. Without rehabilitation, SCI patients may experience muscle weakness, spasticity, contractures, and reduced mobility, which can lead to secondary complications, such as pressure sores, urinary tract infections, and respiratory problems [35]. The patients who did not engage in rehabilitation after SCI had worse outcomes in terms of functional independence and QOL compared to those who received rehabilitation [37]. In addition, another study carried out in Italy indicated that the predominant complications among patients suffering from TSCI were pain, urinary tract infections, lung infections, and bedsores [38]. Therefore, SCI patients must receive timely and appropriate rehabilitation services to maximize their functional recovery and improve long-term outcomes.

This study has several limitations that must be considered. First, this study is a descriptive study that relied on data from hospital records and only represents a small proportion of all individuals with SCI in the region. Additionally, Hubei Province, where this study was conducted, does not currently have a system for registering SCI cases, so the actual incidence rate of SCI in the population could not be accurately determined. Second, the data for this study were collected retrospectively, meaning that the information from past medical records was reviewed. As a result, some loss of data was inevitable due to incomplete or missing records. Third, individuals who died at the accident site or on their way to the hospital were not included in this study, which may have resulted in an underestimation of the prevalence rate. Fourth, the latest International Spinal Cord Injury Core Dataset version (version 3.0) was not used in this study, but instead the medical record version 1.1. It is important to note that this study only included a small sample of traumatic injuries and was conducted in Wuhan, China, so the findings may not be generalizable to other populations. Overall, more research is needed to understand the impact of the COVID-19 pandemic on the incidence rates of SCI in different countries and communities.

## 5. Conclusions

Our study has provided valuable insights into the demographic and clinical characteristics of SCI patients during the non-COVID-19 and COVID-19 periods in Hubei province, China, indicating the need for additional research on the epidemiology of SCI in this province. The results emphasized the need for targeted prevention strategies and appropriate and timely rehabilitation programs to improve the outcomes of SCI patients. Future research should focus on investigating the long-term impact of the COVID-19 pandemic on SCI patients and exploring potential interventions to mitigate the negative consequences of the pandemic on this vulnerable population.

## Figures and Tables

**Figure 1 medicina-59-01699-f001:**
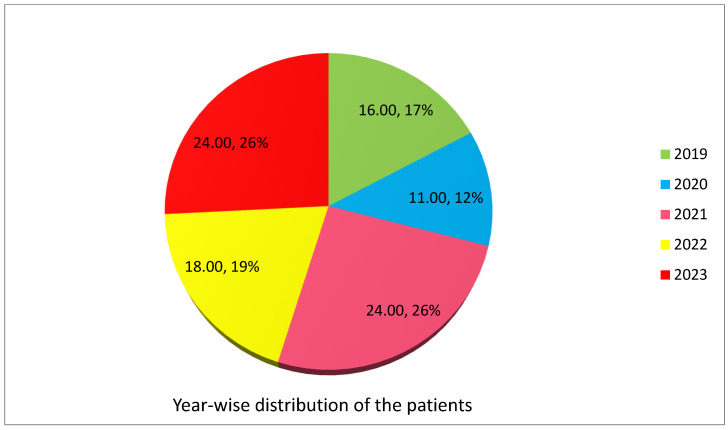
Percentage distribution of patients by year.

**Figure 2 medicina-59-01699-f002:**
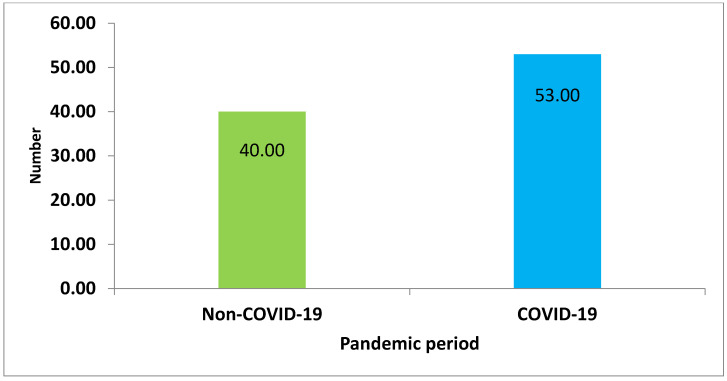
Non COVID-19 period patients (years 2019 and 2023) and COVID-19 period patients (years 2020–2022).

**Figure 3 medicina-59-01699-f003:**
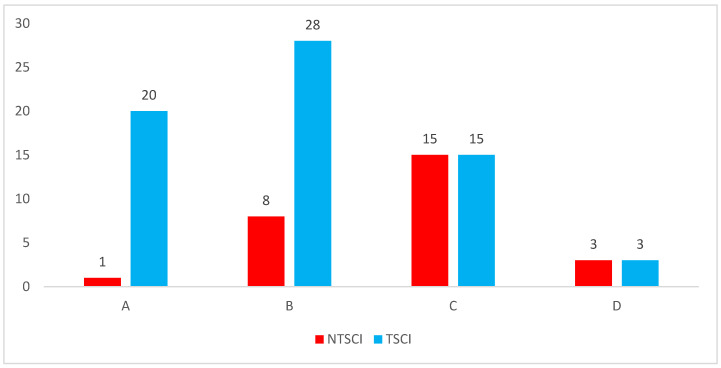
ASIA grades (A, B, C, and D) with traumatic spinal cord injury (TSCI) and non-traumatic spinal cord injury (NTSCI).

**Figure 4 medicina-59-01699-f004:**
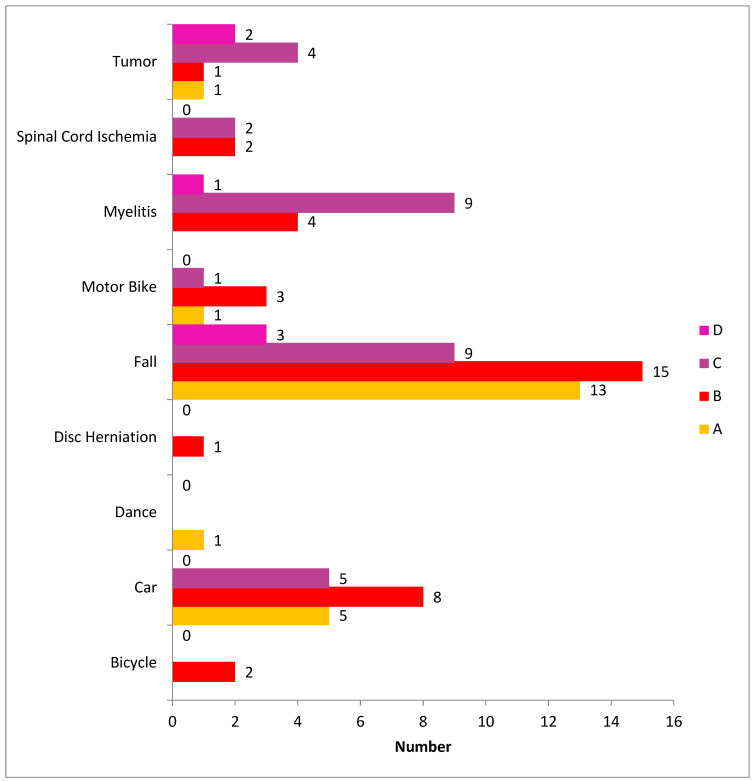
ASIA grades A, B, C, and D with the mechanism of spinal cord injury.

**Figure 5 medicina-59-01699-f005:**
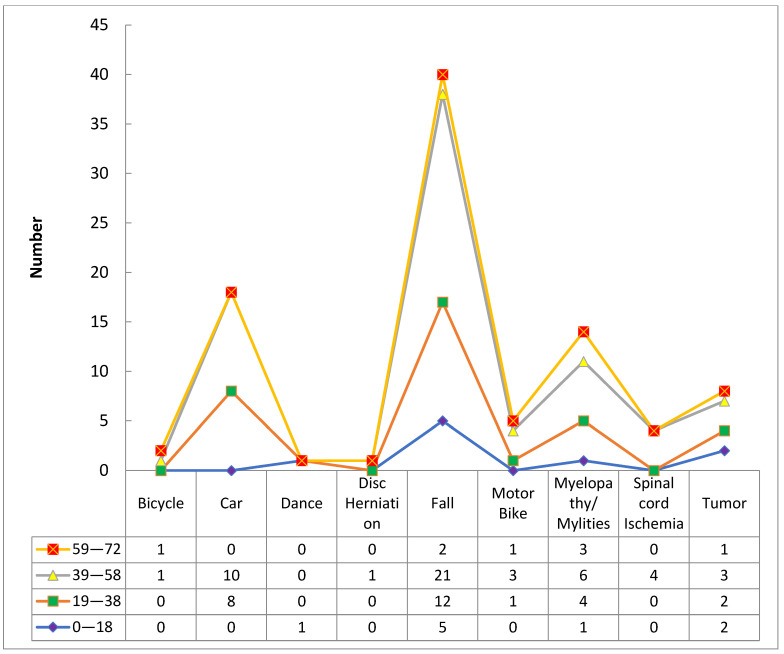
Age with the mechanism of spinal cord injury.

**Figure 6 medicina-59-01699-f006:**
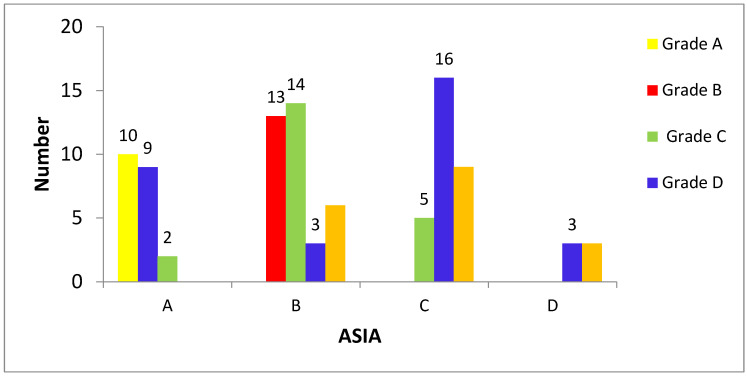
The initial American Spinal Injury Association (ASIA) grades (A, B, C, and D).

**Table 1 medicina-59-01699-t001:** Comparison of the patients’ characteristics in non-COVID-19 period patients (years 2019 and 2023) and COVID-19 period patients (years 2020–2022).

Characteristics		Period		*p*-Value
		Non-COVID-19	COVID-19	
Age (years), mean ± SD		38.80 ± 17.71	44.53 ± 13.27	0.078
Gender, *n* (%)	Male	26 (65%)	35 (66.04%)	1.000
	Female	14 (35%)	18 (33.96%)	
Marital status, *n* (%)	Married	26 (65%)	43 (81.13%)	0.097
	Unmarried	14 (35%)	10 (18.87%)	
Education, *n* (%)	College	8 (20%)	23 (43.4%)	0.186
	High school	10 (25%)	11 (20.75%)	
	Kindergarten	2 (5%)	0 (0)	
	Master’s degree	1 (2.5%)	0 (0)	
	Middle school	8 (20%)	9 (16.98%)	
	No formal education	8 (20%)	7 (13.21%)	
	Primary school	3 (7.5%)	3 (5.66%)	
Occupation, *n* (%)	Driver	3 (7.5%)	3 (5.66%)	0.293
	Farmer	4 (10%)	2 (3.77%)	
	Government office	7 (17.5%)	9 (16.98%)	
	Kindergarten student	2 (5%)	0 (0)	
	Other	7 (17.5%)	8 (15.09%)	
	Retired	1 (2.5%)	2 (3.77%)	
	Student	3 (7.5%)	3 (5.66%)	
	Teacher	1 (2.5%)	10 (18.87%)	
	Worker	12 (30%)	16 (30.19%)	
Ethnicity, *n* (%)	Han	39 (97.5%)	45 (84.91%)	0.051
	Miao	0 (0)	2 (3.77%)	
	Tu Jia	0 (0)	6 (11.32%)	
	Zhuang	1 (2.5%)	0 (0)	
Duration of injury, (Median (IQR))		12 (2.00%)	12 (12.00%)	0.515
Age at the time of injury		37.93 ± 17.76	43.81 ± 13.43	0.072
Type of damage, *n* (%)	Non-traumatic	9 (22.5%)	18 (33.96%)	0.257
	Traumatic	31 (77.5%)	35 (66.04%)	
Initial ASIA, *n* (%)	A	10 (25%)	11 (20.75%)	0.517
	B	18 (45%)	18 (33.96%)	
	C	10 (25%)	20 (37.74%)	
	D	2 (5%)	4 (7.55%)	
Level, *n* (%)	Cervical	22 (55%)	16 (30.19%)	0.004
	Thoracic	9 (22.5%)	21 (39.62%)	
	Lumber	8 (20%)	5 (9.43%)	
	Thoracolumbar	1 (2.5%)	11 (20.75%)	
Paralysis type, *n* (%)	Paraplegia	21 (52.5%)	41 (77.36%)	0.015
	Tetraplegia	19 (47.5%)	12 (22.64%)	
MOI, *n* (%)	Bicycle	1 (2.5%)	1 (1.89%)	0.764
	Car	7 (17.5%)	11 (20.75%)	
	Dance	1 (2.5%)	0 (0)	
	Disc herniation	0 (0)	1 (1.89%)	
	Fall	20 (50%)	20 (37.74%)	
	Motorbike	2 (5%)	3 (5.66%)	
	Myelopathy/myelitis	4 (10%)	10 (18.86%)	
	Spinal cord ischemia	1 (2.5%)	3 (5.66%)	
	Tumor	4 (10%)	4 (7.55%)	
Completeness of lesion, *n* (%)	Incomplete	34 (85%)	45 (84.91%)	
	Complete	6 (15%)	8 (15.09%)	
Treatment, *n* (%)	Conservative	6 (15%)	9 (16.98%)	1.000
	Surgical	34 (85%)	44 (83.02%)	
Final ASIA, *n* (%)	A	5 (12.5%)	5 (9.43%)	0.525
	B	7 (17.5%)	15 (28.3%)	
	C	12 (30%)	9 (16.98%)	
	D	9 (22.5%)	13 (24.53%)	
	E	7 (17.5%)	11 (20.75%)	

**Table 2 medicina-59-01699-t002:** Occupation, education, mechanism of injury, and ASIA with level of injury.

		Level of Injury				*p*-Value
		Cervical	Thoracic	Lumber	Thoracolumbar	
Occupation	Driver	3 (7.89%)	1 (3.33%)	1 (7.69%)	1 (8.33%)	0.115
	Farmer	5 (13.16%)	1 (3.33%)	0 (0)	0 (0)	
	Government office	6 (15.79%)	5 (16.67%)	2 (15.38%)	3 (25%)	
	Kindergarten student	1 (2.63%)	1 (3.33%)	0 (0)	0 (0)	
	Other *	6 (15.79%)	5 (16.67%)	4 (30.77%)	0 (0)	
	Retired	0 (0)	0 (0)	1 (7.69%)	2 (16.67%)	
	Student	0 (0)	3 (10%)	3 (23.08%)	0 (0)	
	Teacher	4 (10.53%)	4 (13.33%)	1 (7.69%)	2 (16.67%)	
	Worker	13 (34.21%)	10 (33.33%)	1 (7.69%)	4 (33.33%)	
Education	College	11 (28.95%)	8 (26.67%)	5 (38.46%)	7 (58.33%)	0.943
	High school	9 (23.68%)	7 (23.33%)	3 (23.08%)	2 (16.67%)	
	Kindergarten	1 (2.63%)	1 (3.33%)	0 (0)	0 (0)	
	Master’s degree	1 (2.63%)	0 (0)	0 (0)	0 (0)	
	Middle school	7 (18.42%)	5 (16.67%)	2 (15.38%)	3 (25%)	
	No formal education	6 (15.79%)	7 (23.33%)	2 (15.38%)	0 (0)	
	Primary school	3 (7.89%)	2 (6.67%)	1 (7.69%)	0 (0)	
Initial ASIA	A	12 (31.58%)	9 (30%)	0 (0)	0 (0)	0.009
	B	17 (44.74%)	5 (16.67%)	8 (61.54%)	6 (50%)	
	C	8 (21.05%)	13 (43.33%)	3 (23.08%)	6 (50%)	
	D	1 (2.63%)	3 (10%)	2 (15.38%)	0 (0)	
MOI	Bicycle	0 (0)	0 (0)	1 (7.69%)	1 (8.33%)	0.596
	Car	11 (28.95%)	6 (20%)	1 (7.69%)	0 (0)	
	Dance	0 (0)	1 (3.33%)	0 (0)	0 (0)	
	Disc herniation	1 (2.63%)	0 (0)	0 (0)	0 (0)	
	Fall	18 (47.37%)	10 (33.33%)	7 (53.85%)	5 (41.67%)	
	Motorbike	2 (5.26%)	2 (6.67%)	0 (0)	1 (8.33%)	
	Myelitis	3 (7.89%)	6 (20%)	2 (15.38%)	3 (25%)	
	Spinal cord ischemia	0 (0)	2 (6.67%)	1 (7.69%)	1 (8.33%)	
	Tumor	3 (7.89%)	3 (10%)	1 (7.69%)	1 (8.33%)	

* Other included unemployed and self-employed individuals.

**Table 3 medicina-59-01699-t003:** Age with the mechanism of spinal cord injury.

MOI	Age Groups (Years)			
	0–18	19–38	39–58	59–72
Bicycle	0 (0)	0 (0)	1 (2.04%)	1 (12.5%)
Car	0 (0)	8 (29.63%)	10 (20.41%)	0 (0)
Dance	1 (11.11%)	0 (0)	0 (0)	0 (0)
Disc herniation	0 (0)	0 (0)	1 (2.04%)	0 (0)
Fall	5 (55.56%)	12 (44.44%)	21 (42.86%)	2 (25%)
Motorbike	0 (0)	1 (3.7%)	3 (6.12%)	1 (12.5%)
Myelitis	1 (11.11%)	4 (14.81%)	6 (12.24%)	3 (37.5%)
Spinal cord ischemia	0 (0)	0 (0)	4 (8.16%)	0 (0)
Tumor	2 (22.22%)	2 (7.41%)	3 (6.12%)	1 (12.5%)

**Table 4 medicina-59-01699-t004:** ASIA scale with gender.

ASIA		Gender		*p*-Value
		Male	Female	
ASIA	A	14 (22.95%)	7 (21.88%)	0.066
	B	24 (39.34%)	12 (37.50%)	
	C	22 (36.07%)	8 (25.00%)	
	D	1 (1.64%)	5 (15.63%)	

**Table 5 medicina-59-01699-t005:** Final ASIA scale with treatment method.

ASIA		Treatment		*p*-Value
		Conservative	Surgical	
Final ASIA	A	1 (6.67%)	9 (11.54%)	0.001 *
	B	1 (6.67%)	21 (26.92%)	
	C	1 (6.67%)	20 (25.64%)	
	D	3 (20.00%)	19 (24.36%)	
	E	9 (60.00%)	9 (11.54%)	

*p*-value *: Significance value.

## Data Availability

The data presented in this study are available upon request from the corresponding author. The data are not publicly available due to patient confidentiality and data privacy regulations.

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
