# Peer review of "The Impact of COVID-19 on Epidemiological Features of Spinal Cord Injury in Wuhan, China: A Comparative Study in Different Time Periods"

_medicina, 2023, doi:10.3390/medicina59101699_

Round 1

Reviewer 1 Report

I would like to congratulate the authors for their work. They have presented a descriptive study among patients with spinal cord injury before and during the covid pandemic, as well as clinical management. Given the significant impact of SCI in the quality of life, I believe the information provided in this manuscript are worthy of archiving.

 Recommendations:

-minor grammatical corrections

minor grammatical corrections

Author Response

Dear reviewer thank you very much for you comments and we appreciate your time to review our article and here is the reply to your comments :

Point 1: minor grammatical corrections

Response 1: Here we modify the grammatical mistakes as fellowing:
Page 1 line 15: This study aims to describe the epidemiological profile of SCI during a period of the pandemic COVID-19 and without 19 COVID-19 in China

Page1 line 19-22: The mean ages were 38.80±17.71 and 44.53±13.27 years, respectively, with, consistent male-to-female ratio of 2:1 across both periods. Notably, falls accounted for the most prevalent mechanism of injury, constituting 50% of cases during the non-COVID-19 period and 37.74% during COVID-19.

Page 1 line 24: Conclusions: A greater

Page 1 line 29: non-COVID-19

Page 2 line 39: cases per million population per year before the pandemic

Page 2 line 48: Rehabilitation can improve motor-sensory function, QOL, and prevent long - term complications such as pressure ulcers and musculoskeletal complications

Page 2 line 50, 51: New research on the use of robotic technology in rehabilitation departments after SCI treatment [7-11], as well as the cooperation of multidisciplinary teams in providing a high-quality care to the SCI patients

Page 2 line 55: discuss the challenges presented by COVID-19 in the context of rehabilitation after SCI.

Page 2 line 58, 59: As far as, there is no community-based registration system for TSCI or NTSCI established in Wuhan city during two different periods (the first one without the COVID-19 outbreak and the other during the COVID-19).

Page 2 line 60: Obtained the patients’ data for this study from the medical records after exact diagnosis with either TSCI or NTSCI according to the Affiliated Tongji College Hospital during a specific time (non-COVID-19 and COVID-19 periods).

Page 2 line 63: non-COVID-19 presented in 2019 and 2023, while COVID-19 in the 2020-2022 year

Page 2 line 70: The patient’s records provided the demographic characteristics, including age, gender, marital status, education level, work type, ethnicity, and the specific mechanism of injury (MOI).

Page 2 line 73: However, the Medical records from December 2019 to April 2023 included information on the injury severity,

Page 3 line 81: The inclusion criteria for the study were SCI patients who admitted to the hospital within the specified time from December 2019 to April 2023 and transferred to the rehabilitation department

Page 3 line 93, 94: machinery - related injuries, sports injuries (dance), and NTSCI are caused by disc herniation, myelopathy, myelitis, spinal cord ischemia, and spinal cord tumors

Page 3 line 102,103: Furthermore, statistical tests, such as chi- square tests

Page 3 line 106-111: From 2019 to 2023, a total of 162 cases of spinal cord injury (SCI) were initially considered for this study. However, during the screening process, 69 cases were excluded for various reasons. These reasons included 22 cases with traumatic brain injury, 16 cases with spinal column fractures but no SCI, 13 cases with heart disease, 15 cases with unclear diagnoses, and 19 cases with incomplete medical records or uncertain diagnoses distribution of patients was 16 patients (17%) in 2019, 11 patients (12%) in 2020, 24 patients (26%) in 2021, 18 patients (19%) in 2022, and 24 patients (26%) in 2023 (Figure 1). (Figure 1). Finally,

Page 3 line 112: 40 patients of non-COVID-19 in the 2019 and 2023 period

Page 3 line 116: 40 SCI people collected from the non-COVID-19 period was 38.80±17.71 years

Page 3 line 117: was mainly high school 10 (25%).

Page 3 line 119: The most common ASIA grade was B,

Page 4 line 121: However, 34 (85%) patients required surgical treatment methods

Page 4 line 124: A total of 53 SCI people were collected from the COVID-19 period according to inclusion and exclusion criteria

Page 4 line 127: Work type was divided into 9 categories which showed the most frequent occupational was a worker

Page 4 line 129: The most common initial ASIA was grade C in 20 patients (37.74%).

Page 4 line 130: However, MOI is divided into 9 categories

Page 4 table 1: Master’s degree

Page 5 table 1: Duration of injury [(Median (IQR)]

Page 5-line 149 Figure 1. Percentage distribution of patients’ year-wise.

Page 6 line 164: Injury to the cervical region formed the first peak followed by the thoracic region in the non-COVID-19 period, while the thoracic region comprises

Page 6 line 166: and thoracolumbar area injuries 13 (34.21%),

Page 7 line 173: the most common grade in NTSCI

Page 7 table 2: Master’s degree

Page 7 table 2: Primary school

Page 8 line 191: There was no statistically significant difference in the distribution of injury levels across different mechanisms of injury (p=0.596) (Table 2).

Page 9-line 209 Figure 4. ASIA with the mechanism of spinal cord injury.

Page 11 line 291: Regarding gender, both groups had a higher proportion of male patient’s male-to-female

Page 112 line 343-345: During the non-COVID-19 period, ASIA grades A and B comprised the majority of SCI cases. During the COVID-19 period, the majority of spinal cord injury (SCI) cases were classified as ASIA B and C injuries in the Asia region

Page 13 line 367: As well as, an association between the ASIA and MOI there was a statistically significant

Reviewer 2 Report

The Manuscript “The Impact of COVID-19 on Epidemiological features of spinal cord injury in Wuhan, China: A Comparative Study in Different Time Periods.” is an original article.

The author has taken the case history of spinal cord-injured patients during covid 19 and non-covid 19 periods. It is a retrospective study.

In general, the manuscript is well written. This research has been designed in an appropriate manner. There are a few suggestions to improve the manuscript.

Page number 1 - line number 20 The sentence needs to be rephrased. There is a lack of clarity and incompleteness in the statement.

Page No. 2 – Line No. 67 The sentence has to be rephrased. It is not clear and incomplete. The time period specified is unclear. Please specify the duration of each period appropriately, including the month, year, and duration of each period.

Page No. 2 – Line No. 71 The sentence is incomplete. It is without any meaning.

Page No. 2 – Line No. 83 I would like to sumptuous the sentence. The latest International Spinal Cord Injury Core Data Set version (version 3.0) should have been used for this study. I would appreciate it if you could add to the limitations or justify the version used.

Page No. 3 – Line No. 99-106 is not clear. It might be better to rephrase these sentences to make them easier for the readers to understand.

Page No. 12 – Line No. 344 Rephrase the sentence.

Since the study is concerned with the pandemic period, data might be collected from different countries in order to validate the study in an efficient and effective manner. The same can be included in the scope/limitations of the study,

Moderate English corrections is mandate.

Author Response

Dear reviewer we appreciate your time to reviewer our paper and here is the reply to your all comments:

Point 1: Page number 1 - line number 20 The sentence needs to be rephrased. There is a lack of clarity and incompleteness in the statement.

Response 1 line 20: The sentence mentioned is rephrased to: The mean ages were 38.80±17.71 and 44.53±13.27 years, respectively, with, consistent male-to-female ratio of 2:1across both period

Point 2: Page No. 2 – Line No. 67 The sentence has to be rephrased. It is not clear and incomplete. The time period specified is unclear. Please specify the duration of each period appropriately, including the month, year, and duration of each period.
Response 2: The sentence in line No. 73 became like this: However, the Medical records from December 2019 to April 2023

Point 3: Page No. 2 – Line No. 71 The sentence is incomplete. It is without any meaning.
Response 3: The sentence in line No. 81 became like this: The inclusion criteria for the study were SCI patients who admitted to the hospital

Point 4: Page No. 2 – Line No. 83 I would like to sumptuous the sentence. The latest International Spinal Cord Injury Core Data Set version (version 3.0) should have been used for this study. I would appreciate it if you could add to the limitations or justify the version used.

Response 4: The sentence is added to the limitation in line No. 410-412

Point 5: Page No. 3 – Line No. 99-106 is not clear. It might be better to rephrase these sentences to make them easier for the readers to understand.
Response 5: The sentence in lines 106-111 rephrased to: From 2019 to 2023, a total of 162 cases of spinal cord injury (SCI) were initially considered for this study. However, during the screening process, 69 cases were excluded for various reasons. These reasons included 22 cases with traumatic brain injury, 16 cases with spinal column fractures but no SCI, 13 cases with heart disease, 15 cases with unclear diagnoses, and 19 cases with incomplete medical records or uncertain diagnoses.

Point 7: Since the study is concerned with the pandemic period, data might be collected from different countries in order to validate the study in an efficient and effective manner. The same can be included in the scope/limitations of the study,

Reviewer 3 Report

In this manuscript, the authors investigated the epidemiological features of spinal cord injury (SCI) in a period of pandemic COVID-19 and without COVID-19 in Wuhan, China. The authors found that a greater proportion of males suffer from SCI, and the primary causes are falls and traffic accidents. They also mentioned that workers are the most vulnerable group to SCI among all patients. This study focuses on the importance of SCI rehabilitation. The study is interesting and well-written. However, the authors need a revision before acceptance for publication, as follows:

(1)   Figure 2: Write the full form of “N” in y-axis title.

(2)   Missing of y-axis title in Figure 2.

(3)   Please mention the full form of different groups (A, B, C, and D) in Figure 3 and Figure 6 legends. Mention the full forms of TSCI and NTSCI in Figure 3 legend.

(4)   Correct the part of sentence “Spinal cord injury (SCI) SCI” (page 1, line 13) as well as “without 19 COVID-19 in China” (page 1, line 13).

(5)   Mention the full forms of TSCI and NTSCI (page 2, line 60).

(6)   Please correct the grammatical error for the sentence “Moreover, in this study used the International Spinal Cord Injury Core Data Set version 1.1” (page 2, line 82-83).

(7)   Briefly described the classification of American Spinal Injury Association (ASIA) scale (A, B, C, D, E, F) as well as the specific mechanism of injury (MOI) with references in the Method section.

(8)   Materials and Methods: Put the “2.3. Ethical considerations” sub-section” before the “2.1. Data collection and statistical analysis” sub-section.

Author Response

Point 1: Figure 2: Write the full form of “N” in y-axis title.

Response 1:

Point 2: Missing of y-axis title in Figure 2.

Response 2:

Point 3: Please mention the full form of different groups (A, B, C, and D) in Figure 3 and Figure 6 legends. Mention the full forms of TSCI and NTSCI in Figure 3 legend.

Response 3:

Figure 3. ASIA scale with traumatic spinal cord injury (TSCI) and non-traumatic spinal cord injury (NTSCI).

Figure 6. Initial and final American Spinal Injury Association (A, B, C, D, E) grades.

Point 4: Correct the part of sentence “Spinal cord injury (SCI) SCI” (page 1, line 13) as well as “without 19 COVID-19 in China” (page 1, line 13).
Response 4 line 11, 15: Spinal cord injury (SCI) is a severe affliction that can have a profound impact on a person's ability to move and feel, affecting a significant number of individuals.
This study aims to describe the epidemiological profile of SCI in a period of pandemic COVID-19 and without COVID-19 in China,

Point 5: Mention the full forms of TSCI and NTSCI (page 2, line 60).
Response 5 line 64: traumatic spinal cord injury (TSCI) and non-traumatic spinal cord injury (NTSCI).

Point 6: Please correct the grammatical error for the sentence “Moreover, in this study used the International Spinal Cord Injury Core Data Set version 1.1” (page 2, line 82-83).
Response 6: The mentioned sentence changed to: Moreover, in this study we have used the International Spinal Cord Injury Core Data Set version 1.1 line No. 79

Point 7: Briefly described the classification of American Spinal Injury Association (ASIA) scale (A, B, C, D, E, F) as well as the specific mechanism of injury (MOI) with references in the Method section.
Response 7 in line 73, 77: traumatic (vehicle accidents, fall, sport injury) non traumatic (disc Herniation, myelopathy/ myelitis, spinal cord ischemia, and tumor) [13]. However, the patients record included information on the injury severity, as measured by the American Spinal Injury Association (ASIA) is a standardized test that evaluates the extent of spinal cord injuries. It assesses motor function based on specific muscle groups, sensory function based on specific areas of the body, and also includes an examination of the anus and rectum. The results of these tests determine the severity and location of the injury [14].

Point 8: Materials and Methods: Put the “2.3. Ethical considerations” sub-section” before the “2.1. Data collection and statistical analysis” sub-section.

Response 8 line 65: 2.1. Ethical considerations
  This study was approved by the ethical committee of the Affiliated Tongji Medical College Hospital (Ethical approval number: TJ-IRB20210314). 
